# Optimizing Operations on B-Trees Using Proximal Policy Optimization and Hierarchical Attention-Based Models

## Abstract

Modern database management systems often rely on B-trees to achieve indexing in an efficient manner. If stored on slow permanent storage devices, write and read operations can become a significant performance factor, as transactional databases require regular additions and deletions. We propose to use a reinforcement learning setup to optimize the write performance of deletes and inserts by aggregating them and optimizing their order of execution. This achieves the goal of minimizing write times during tree updates. We present a small hierarchical attention-based model to parse the content of the tree efficiently. The new architecture allows for level-wise parallel computation and includes caching to improve the inference speed. Our evaluation verifies the applicability and the potential of the proposed framework. We show that we can efficiently compute an embedding in a hierarchical dataset and that the embedding can be used to achieve noticeable performance improvements in B-tree operation scheduling in comparison to accepting operations in their order of arrival.

## 1 Introduction

B-trees are a well-known data structure in the database research community. They are commonly used to index data and make it searchable efficiently. Inserting and deleting new data in a B-tree and then reorganizing the tree is nontrivial. In some practical applications, those trees can become very large, containing millions of entries. If insert and delete operations occur at high frequency throughout the tree's existence, they lead to regular changes to the tree's structure. Those become crucial if, throughout the insertion, a node overflows and needs to be split or is underfilled, in which case they need to be merged. Depending on the tree implementation, noticeable overall performance improvements could be achieved by collecting received operations and reordering them to minimize the overall cost of execution. While elegantly designed heuristics might achieve a comparable goal, those probably require full access to the tree during each optimization step to find an efficient solution. We propose to learn a representation of the tree that can be efficiently computed during updates and use this to automatically train an agent to choose the order of execution. This approach aims to be scalable to large tree sizes and many operations, but will be evaluated on small setups that allow tractable analysis of the optimal solution.

To further motivate this concept, our system setup is shaped by the following characteristics: 1. The indexed data and the tree itself are stored in permanent storage with high-latency access in comparison to the memory of the machine executing the operations. Write operations are especially costly, while sequential reads are, due to the nature of intelligent prefetching and caching on modern hardware, the most efficient operation on those storage systems. 2. During an insert/delete, costly rebalancing operations should be avoided. Thus, the inserts and deletes should occur in a way that minimizes the need for rebalancing later on.

A B-tree forms a search tree with nodes of variable size. It is characterized by the maximum number of referenced child nodes within a single node, a number called $b$ throughout this paper. To define the borders between $b$ nodes, $b - 1$ keys are required. Every referenced node shall only contain values between the two keys surrounding the value. To avoid large unfilled nodes, the nodes have a minimum width, which is set to $\lceil 0.5b \rceil$. To stay within these boundaries, inserts and deletes cause

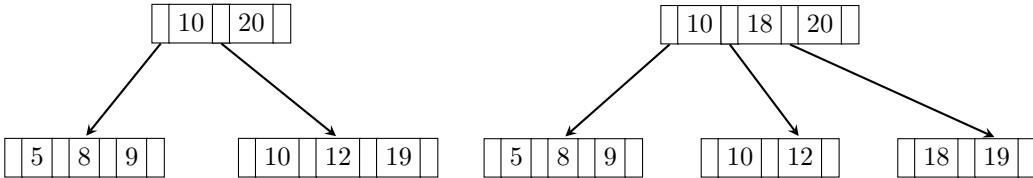

Figure 1: The resulting tree when an already full node on a $b = 3$ tree would require an additional child at key $n = 18$. The outermost node is split, and at least three nodes need to be partly rewritten. This propagates to the parent and could cause additional reorderings if the parent node exceeds its maximum size after executing this operation.

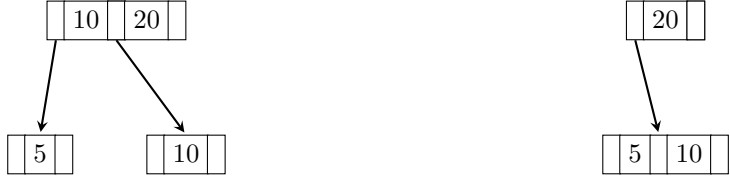

Figure 2: The resulting tree when two neighboring nodes are merged due to both being underfilled, in this case having less than two keys. Again, the resulting write operation causes at least two nodes to be rewritten in storage.

reordering of nodes if they are exceeded after insertion or deletion. An illustration of the insert- and delete process with overflowing and underfull nodes is given in Figure 1 and Figure 2. The illustration omits any changes outside of the three nodes referenced, while in practice, a split or merge might occur more than once while traversing the tree towards the position where the element has to be inserted or removed.

**Contribution**    To circumvent the challenges above, we seek to design a network architecture with an efficient caching mechanism to minimize the number of reads and writes on the data structure while parsing and throughout operation. Our contributions can be summarized as follows:

- We propose a framework to efficiently parse large hierarchical datasets, which iterates the tree for evaluation.

- Our approach incorporates a novel caching mechanism that makes use of that data structure.

- We use that model and the caching mechanism in conjunction with Proximal Policy Optimization to learn a prioritization policy to dynamically manage write and delete operations on B-trees.

- We provide numerical examples and performance comparisons against chosen heuristics, as well as the optimal achievable performance.

- We provide an open-source simulation and evaluation toolkit as a public code repository.

Throughout the next sections, we show that an agent can be trained based only on interaction with the system and that its policy outperforms other plausible heuristics that could be applied to the same problem. At the same time, once trained and under careful optimization to minimize necessary tree reads, it is quick enough not to diminish the performance improvements it achieves during read/write operations. Section 2 introduces the domain of B-trees as well as comparable architectures. Section 3 introduces the model architecture. Section 4 provides initial regression experiments. Section 5 introduces the way the Markov process is shaped that our toolkit will solve. Section 6 compares the overall system to different baseline approaches as well as different model setups. Section 7 concludes.

## 2 RELATED WORK

An overview of operation optimization on index structures is given in the following Section 2.1. Our model is highly inspired by recurrent neural networks and transformer architectures. Thus, Section 2.2 provides an introduction to comparable systems.

### 2.1 B-TREE AND ITS OPTIMIZATIONS

B-trees are a data structure long known in database and algorithm literature (Sedgewick & Wayne, 2011). They are used to index large, unsorted data structures. A B-tree offers a higher width per node than binary search trees. This allows the node to directly point to several children, which allows a tree format with a low height and a node block size that can be optimized for read/write performance on lower-tier storage devices.

There are several variants that further improve B-trees, namely B* trees and B+ trees. For example, a B+ tree distinguishes itself from a B-tree through the fact that B+ trees allow efficient sequential search by storing a pointer to the next sibling within each leaf (Comer, 1979).

A long track of research exists concerning optimization of the storage of a tree within different storage tiers, like optimizing cache hits when reading large trees from storage, for example, in Rao & Ross (2000). Others aim at optimizing them for nonvolatile memory instead of relying on traditional storage devices (Chen & Jin, 2015). None of the previously mentioned studies aims directly at optimizing the insertion process itself. Closely related are those that do optimize the internal process of insertion or deletion, as described in Jannink (1995). Optimizations in that regard could theoretically be used in conjunction with our approach. In this setup, we could think of one additional optimization: When multiple values need to be inserted or deleted, several operations can be performed in bulk, and unless a node overflows, the rebalancing is postponed until the batch operation is evaluated. As this might lead to overflows or empty nodes if executed wrongly, the use case for our approach still exists if a prediction can be performed fast enough.

B-trees can be accessed in parallel. Native write access to such a tree requires locks and thus, possibly forces sequential execution if locks collide. To overcome this limitation, lock-free alternatives would be of interest, as described in Braginsky & Petrank (2012). If bandwidth is limited, prioritization might still be necessary, which gives space for input order optimization as described here.

Kraska et al. (2018) started a fundamental discussion on the suitability of learned indices compared to classical index structures, including B-trees. While replacing the whole structure would render any optimization on it obsolete, we assume that the B-tree will still be in use in diverse environments. A different key takeaway, which is highly relevant to our work, consists of the proof that low-level operations like traversing a B-tree can be replaced by a model in some cases. In those cases in which fully replacing the index structure is not feasible, our architecture can still support the auxiliary processes around the index when needed.

### 2.2 TREE NEURAL NETWORKS

Parsing sequential inputs is a well-discussed challenge in machine learning research. Up to the rise of transformers, architectures like Recurrent Neural Networks (RNNs) have been used for that purpose, their earliest descriptions are provided in Rumelhart et al. (1986). In our scenario, the input is not a sequence, but a tree instead. Further architectural developments led to LSTMs, a seemingly superior architecture for parsing sequences (Hochreiter & Schmidhuber, 1997).

A related, node-oriented architecture is described in Ren et al. (2021). In their research, a tree was parsed top-down, with the child nodes receiving the parent nodes' output if the respective node was chosen by a classifier to be the correct element. Cheng et al. (2018) propose to use a comparable architecture that parses each node of the tree by using the values of the left sibling as well as the outermost right child of the current node as input to each iteration step. They target the problem domain of evaluating sentences that are already preprocessed in a tree structure. The horizontal relationship hinders caching mechanisms, such as the one described here. Most of those architectures have been superseded by the transformer, as outlined by Vaswani et al. (2017). The attention layer of a transformer was reused in the hierarchical encoder described in Section 3.

## 3 REPRESENTATION MODEL SETUP

The network needs access to the data within the tree to perform its evaluation. The architecture proposed in the next section aims to reduce the effect of this limitation. This is the case for both the policy and the value network, as both use the complete state of the system as their input. The first layers of both networks are shared and contain our encoder setup. The network is optimized using ADAM (Kingma & Ba, 2015), a brief overview of the most important hyperparameters is given in Table 2 in Appendix E.

### 3.1 HIERARCHICAL MODEL ARCHITECTURE

A challenge of this and other related problems is induced by its input. Our solution aims to efficiently embed the whole B-tree at inference time in a live application. It lies in the nature of B-trees that they can become very large if they are used to index large columns in database management systems. Flattening the whole tree and passing it to a large network is inefficient, or in many cases, even intractable due to the increasing number of neurons required in the first layer, as shown in Table 4.

We choose to tackle this challenge by embedding the B-tree recursively, to keep the number of trainable parameters constant while increasing tree size. A full visual description of our hierarchical architecture is displayed in Figure 3. The following steps compute the embedding:

- We embed the leaf values using a linear projection layer to reshape them to the same dimension as the encoder, so it can be used to parse all nodes.

- For the internal nodes, we concatenate all child embeddings and the node keys before feeding them into a feedforward layer.

- To improve performance, we add a positional embedding based on the depth of the node currently being parsed. The positional encoding is a unique learnable parameter for every layer of the tree.

- The aggregated child embeddings are then passed to a transformer encoder, which computes the embedding of the node currently under assessment. The attention layer follows the classical layout described by Vaswani et al. (2017):

$$f(Q, K, V) = \text{softmax}\left(\frac{QK^T}{\sqrt{d_k}}\right)V. \tag{1}$$

  The transformer consists of one layer of self-attention, followed by two linear layers as part of the feedforward network, though other setups were validated as well. The dimension of a node embedding in our system is usually 64. A setup replacing the attention layer with a linear layer was also tested, and the results are depicted in Figure 5.

Our hierarchical encoder allows us to process nodes within one tree layer in parallel, as each node is only dependent on its child embeddings and its own keys. The final root embedding is then passed to the value and policy network of the agent, which consists of two linear layers followed by a non-linear activation function. The resulting values are then either compressed to a single value estimation in the case of the value network or a vector of the same size as the theoretically available action space, and then normalized to form a valid distribution over actions. Each action that is masked out receives a probability of zero at this point.

### 3.2 CACHING

The proposed structure allows for further optimization. Many operations change only small or even single elements of the tree. The embeddings of all unchanged subtrees below or parallel to the changed node stay the same and thus do not need to be recomputed during operation. If an element is changed, the embeddings of all of its children can be reused, as well as all other nodes except those on the path of this element to the root.

The invalidation bit of every node is saved in an extra array, which resembles the node structure of the tree and is small enough to stay in the main memory. It can thus be accessed quickly. If a node is changed throughout the execution of the operation, it is invalidated and its embedding will be

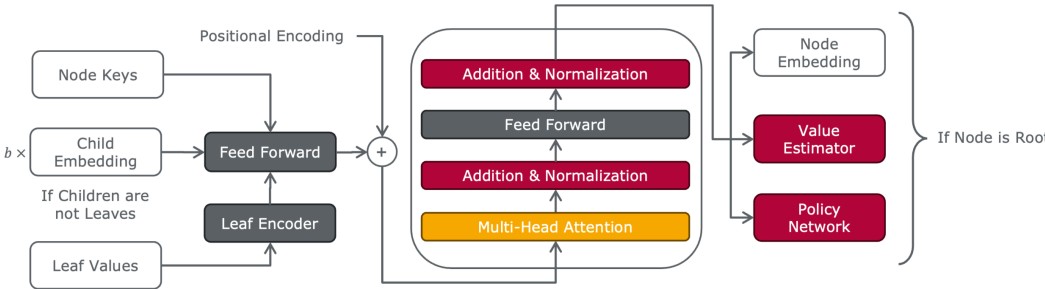

Figure 3: A visualization of our hierarchical embedding setup. A node's embeddings can be calculated using only its keys and its $b$ children. The resulting values are either propagated upward or passed to the value and policy network once the root of the tree is reached. The output of each node is cached and made available to parents from the cache if it is available and unchanged.

recomputed through the next cycle of the forward pass. As displayed in Figure 7, all nodes along its path to the root need to be invalidated as well. The root node will always be invalidated, so the final embedding has to be computed at all times.

The embeddings themselves should be significantly smaller than the actual node size for the effect of caching to be maximized. Even if the embeddings are of comparable size, the fact that a noticeable number of embedded nodes do not need to be recomputed is still advantageous. Embeddings of the upmost layers of the tree should be kept in memory at all times, as those nodes cover the majority of the nodes within a tree, and a usual update invalidates only one node at the top. If read from storage, the caching effects decrease, but can still be positive if reading a relevant segment of the tree can be avoided.

During training, accessing all embeddings for backpropagation is still necessary in our current implementation. It is highly desirable to use an already optimized policy during deployment and pre-train the agent asynchronously on the dataset. Then, the impact of caching can be put to full effect, as no backward passes need to be performed. Further, the embedding of the tree can be computed after the changes are applied asynchronously, while the agent receives new operations to order.

## 4 REGRESSION EXPERIMENTS

To evaluate the general capabilities of the architecture, we performed several experiments on B-trees of growing sizes on a simpler regression task instead of the more complicated reinforcement learning (RL) setup. The regression target was to predict the cost of executing a specified operation on a given tree, i.e., the reward function of the process described in Section 5.

We pre-generated a set of 1,000,000 diverse trees to allow faster loading. Each tree was set up with four values per node to ensure that split and merge operations appear reliably and create a more diverse reward signal. The final regression setup included 124 values in the leaves of the tree, resulting in a B-tree with 31 to 62 nodes and a depth of 4. Figure 4 displays the performance of our setup (HE w/ Att.) in comparison to a model where we flatten the tree and parse it with a linear layer instead (Flattened w/ MLP). Our architecture successfully predicts the cost associated with inserts and deletes up to this tree size, with larger sizes requiring further evaluation, as the model configuration needs to be changed to fit the complexity and data density of extremely large setups, while not overfitting. The dataset was split into 80% training and 20% evaluation data, and the evaluation data had a mean of 0.25. The low mean is caused by the fact that most B-tree operations do not lead to reordering. The highest observed target value in the data is 3. Further evaluation of our results can be found in Appendix B.

## 5 PROCESS MODEL AND RL SETUP

The following section introduces the general setup of the Markov Decision Process (MDP) that the agent is facing. As an explicit tractable algorithm to determine the value function and the optimal

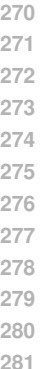

Figure 4: Comparison of the evaluation loss throughout the training runs with our hierarchical attention setup (HE w/ Att.), against a linear feature extractor that parses the flattened B-tree (Flattened w/ MLP). Always predicting the mean of the evaluation dataset would lead to an MSE of 0.37.

policy is not known, we search for an optimal policy through reinforcement learning. We will follow the three main components of an MDP: The state $s_t \in S$, the action $a_t \in A$, and the reward $r_t$.

During training, the process is initialized by starting with a randomly generated valid B-tree filled with data and a set of items to append or delete from that tree. The agent then faces the task of determining an order of insertion by selecting an item to either append or delete from that list. After each selection, the operation is carried out without further interaction with the agent in between.

The decision is performed by an agent whose policy is optimized using Proximal Policy Optimization (PPO) (Schulman et al., 2017). PPO relies on the policy gradient theorem, expanded by clipping the policy gradients according to the loss $L$:

$$L(\phi) = \mathbb{E}_t[\min(r_t(\phi)A_t, \text{clip}(r_t(\phi), 1 - \epsilon, 1 + \epsilon)A_t)]. \tag{2}$$

The agent observes the system and relies solely on observations to optimize its policy. This allows for various distributed loads on the B-tree without readjusting the heuristic used to optimize the order of insertions. The agent relies on two internal models to implement this: The value and the policy network. In our use case, the value network should represent the minimal discounted cost of all open insertions and deletions for the given tree.

**State Formatting**   The state needs to incorporate two pieces of information: The current state of the B-tree, as well as the list of items to insert and delete. The precise state description includes the following: The whole tree under assessment, parsed through our hierarchical network, and the list of elements to insert and delete in an array of fixed length.

**Action Representation and Transition**   The action choice consists of selecting which of the above-mentioned inserts and deletes has to be executed next. The current list of operations to be executed is provided as part of the state description, as mentioned earlier. The action space is designed analogously to the list of operations, with one action corresponding to one element in the input vector.

Action masking (Huang & Ontañón, 2022) is used to dynamically modify the action space to remove all actions that resemble empty spots in the input. In our evaluation setup, this results in fewer actions becoming available throughout the execution of the process.

**Reward Shaping**   The reward is based on the execution costs of each operation performed:

$$r_t = -p \cdot (n_{\text{leafsplit}} + n_{\text{leafmerge}}) - q \cdot (n_{\text{split}} - n_{\text{merge}}). \tag{3}$$

The cost of any operation is determined by the number of nodes split $n_{\text{split}}$ and $n_{\text{leafsplit}}$, as well as merges $n_{\text{merge}}$, $n_{\text{leafmerge}}$ required to successfully execute that operation. The minimum value of a state thus becomes the minimal discounted number of splits and merges required to insert and delete all values given for that specific state. We weighted leaf operations higher than other node operations. Determining the value function of a batch of operations is difficult, leading to an MDP-based model, solved with RL, as the most promising alternative.

To allow a potentially endless running process, the discount value $\gamma$ is set to a value smaller than one. In setups where no further operations are added to the agent's input, a value of one would be theoretically feasible. This setup incentivizes the agent to perform lower-cost operations early in the process. If the value were set to $\gamma > 1$, the agent would be incentivized to execute costly operations as early as possible. A non-discounting setup would void this effect. Future optimization that uses this effect would be possible.

## 6 EVALUATION

We measure the success of the approach by comparing our heuristic to the baseline in terms of the highest achieved reward. The actual overall execution time of read and write operations could be estimated by further measurements. This does not fully represent caching effects when the tree is stored in permanent storage, but it serves as an indicator of real-world performance. The goal of the evaluation is to display that our agent outperforms plausible, quickly computable heuristics and is noticeably faster than exhaustive search, leading to optimal execution order. We show that using a tool like this can significantly reduce the number of operations executed throughout a process. Furthermore, several setups are compared, while several baselines are discussed. As an assistant for future applications, the training performance observed throughout the experiment provides insights into the requirements of the system during training.

### 6.1 BASELINES

We compare our system to one fundamental baseline, which marks the lower bound for any heuristic. It is formed by a random order of operation for a given set of inserts and deletes. This resembles executing them in order if received in random order on a real-world system. Furthermore, we evaluated three non-optimal heuristics. The first one, named alternating in the following text, chooses the largest available items in an alternating fashion. Thus, the largest element from the list of available inserts is chosen, and then the largest element is deleted. This is repeated until the list of available deletes and inserts is empty. The second evaluated heuristic, called insert first, first executes all inserts, followed by all deletes in descending order. The third heuristic, called delete first, executes all deletes in ascending order, followed by all inserts. The average reward of random execution and all heuristics in our evaluation setup is displayed in Table 1.

### 6.2 EVALUATION SETUP

The experiment setup is chosen so that the resulting problems can still be analyzed by a human operator. This setup allows a case-by-case evaluation of our policy in comparison to the heuristics. To achieve this goal, we choose the node size and count to be relatively small. In practice, larger node sizes would be used to maximize write and read performance. Small node sizes also increase the probability of reordering and thus decrease the sparsity of the feedback signal. The sparse operations on larger trees can become much more costly in comparison, as more data needs to be reorganized.

For our evaluation, we also compare the performance of a flattened observation space with a linear feature extractor (Flattened w/ MLP) and an attention-based encoder (Flattened w/ Att.) with our two hierarchical feature extractor variants: Multi-head attention layer after the child combination (HE w/ Att.) and linear layer after child combination (HE w/ MLP). Despite the undesirable scaling properties of flattened feature extractors, we still wanted to compare architectures on small trees. The goal was to determine if our hierarchical architecture could compete on trees where all values can be directly parsed into the feature extractor. With these goals in mind, the parameters of the environment for all following evaluations, if not further specified, are described in Table 3 in the Appendix.

Each run lasted for $10^8$ episodes. Each episode was executed starting with a randomly generated valid tree with 24 values, as well as six randomly generated inserts and six randomly generated deletes. The average reward throughout the training process is displayed in Figure 5. As displayed, our attention-based setup outperforms all other architectures. The average reward of the random policy is -4.76, while the final learning state of the hierarchical feature extractor achieves -2.17, leading to an average improvement of 55% in comparison to randomly selecting elements. We measured the inference time for different setups to ensure the practical applicability of the model, see Appendix H.

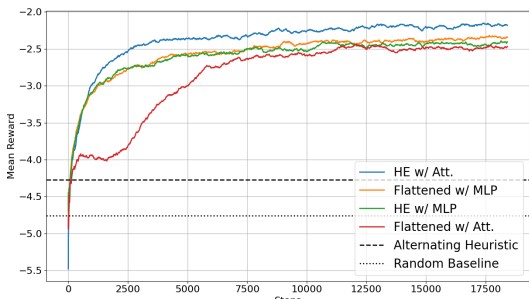

Figure 5: Comparison of the average reward throughout the training runs with our hierarchical attention setup (HE w/ Att.), a hierarchical feature extractor using a feedforward network (HE w/ MLP), and a linear feature extractor that parses the flattened B-tree (Flattened w/ MLP). The fourth setup consists of a model with one attention layer, which receives the flattened tree as input (Flattened w/ Att.). The average rewards of all the heuristics described are shown as horizontal lines.

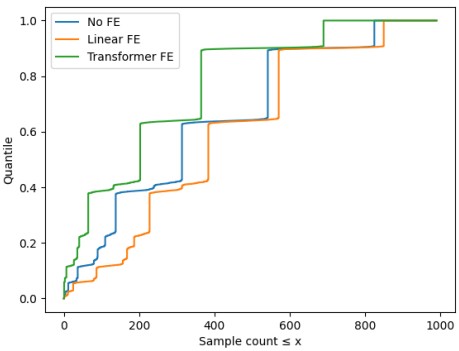

Figure 6: The quantile position of the agent's policy compared to the overall space of possible permutations computed by exhaustive search over 1 000 randomly generated trees. All possible execution orders were aggregated and their reward measured. The best-performing setup achieves a position in the 90% quantile in over 700 of the 1 000 experiments. The steps in the graph are caused by the stepwise reward function of the environment, in which many permutations of operations end up with the same number of overall operations and thus reward. By only improving those by a single reorder a noticeable number of alternative execution orders is outperformed.

A more detailed qualitative analysis was performed to illustrate the space of reachable policies and their average rewards. Figure 6 displays the distribution of quantiles achieved by the agent's chosen solution. The quantiles were computed by estimating the cost of 1,000 permutations of the operations. This process was repeated for 1,000 initial trees. It shows that the system finds a well-performing policy in most of the cases, an illustration of one of those scenarios is given in Figure 9, see Appendix E. A few tree setups still lead to poor resulting performance, as is the case in the example discussed in Figure 10. The hierarchical attention-based setup outperforms all other setups we evaluated for this specific use case. Flattening and using the tree as input yields comparable results, though consistently lower. As discussed earlier, flattening large trees is not viable and only works in an artificially small setup, while the best-performing solutions are, in theory, not limited.

Additionally, we validated the ability of the agent to generalize to unknown tree sizes. While increasing the number of values in our tree to 124, the policy trained on trees with size 24, still outperforms the random and the alternating baseline. After increasing the number of values to 1024, we could not detect a relevant performance improvement over the alternating baseline. The decrease in relative performance might be due to the positional embedding, which encounters unknown tree depths during inference time. Interestingly, this can still be compensated for when only one additional tree level is added while increasing the tree size from 24 to 124. If this trend holds for larger trees, production systems would only need to be fine-tuned after at least a B-fold increase in values.

## 6.3 STORAGE SPACE

We assume that in an applied system, each node requires the full storage space available for the node, independent of the actual number of values in the node. Yao (1978) have shown that if a tree is filled randomly, in theory, 31% of the keys and pointers stay unused. This leads to the question of whether our approach, by minimizing splits, subsequently also increases the storage efficiency. We can show that by using our method, the average storage demand can be temporarily improved by up to 1.06 nodes, or 11%, at the point of maximum difference. This makes this optimization interesting even in those scenarios in which the write/read performance is negligible for the user. While the agent deconstructs this advantage throughout each process, he achieves a noticeably lower result at the end of each 12-step episode. A precise description of the experiment and figures displaying the agent's behavior are provided in Appendix G.

## 6.4 ALGORITHM DISCOVERY AND NO TREE SOLUTION

This project has its foundations in a research project with the goal of searching for fast-performing algorithms for algorithmic problems. Because of this, we performed a qualitative analysis of the outputs of the system for patterns that could, in turn, be developed into a static algorithm that does not require network inference. A brief overview of a set of examples is provided in Figure 13, see Appendix I. It displays that no human observable execution pattern can be discovered when only the chosen actions are considered. To find an algorithm that does not need the tree to function and might thus be observable in the policy, we retrained the agent without any access to the actual tree itself. In this setup, the agent, using only a small feedforward network, observes only the operations to execute and should deduce in which order those operations should be executed to maximize performance.

Figure 12 displays the results. The resulting policy, at least when provided with a sorted input array, achieved surprisingly good performance, -3.1 in the average case. This outperforms any heuristic that we previously evaluated. Accessing the tree does not impede the computation time of the agent, so the extra time spent on computation for the large model might be well invested on slow storage devices when applied to large trees. A qualitative analysis, illustrated in Figure 13, cf. Appendix I, of this agent's policy, again did not reveal any human-readable patterns. Future research could aim at distilling those policies into quickly executable algorithms.

## 7 CONCLUSION

We discovered a model setup that can successfully learn well-performing policies in the provided environment. It is capable of outperforming the execution cost of random execution, as well as any rule-based approach we came up with. We have shown that even without knowledge of the tree, the agent learns a policy that performs well without any necessity to parse the input tree.

To achieve optimal results, our agent must effectively parse the input tree. Our hierarchical encoder structure accomplishes this by parsing from the leaves up to the root. We demonstrated that this novel hierarchical approach outperforms all tested alternatives while remaining agnostic to the tree depth and node structure. Additionally, our evaluations highlight its potential for storage optimization, requiring fewer node splits and merges. Furthermore, we showcase the architecture's potential, including a high caching rate during normal operations and an efficient embedding size.

For future work, testing with upscaled B-trees on an even larger dataset can further illustrate the practical applicability of our methodology. All our setups relied on a dense reward signal, which would become sparse if the node size was increased noticeably. This is expected to occur because larger nodes reach their upper and lower limits less regularly. Evaluating the impact of this effect would provide further insight into its performance on very large datasets.

As our model is agnostic to the problem it is applied to, exploring other applications in which the input data can be reshaped would display future application possibilities. This would apply to all use cases in which the data can be augmented by ordering it in a semantic hierarchy.

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

## A CACHING

Caching allows reusing the embeddings of large elements of the embedding tree. Figure 7 illustrates this potential by displaying the valid and invalid nodes on a small tree and the invalidation path.

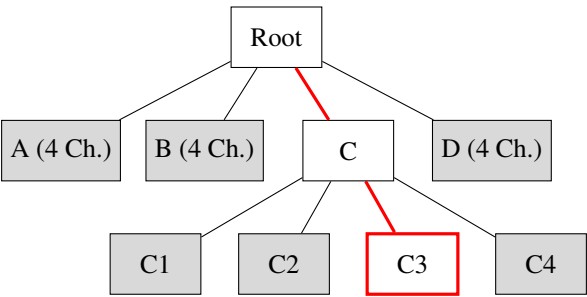

Figure 7: An illustration of the caching mechanism: A change in C3 invalidates all nodes along its path to the root. All gray nodes are read from the cache and do not need to be recomputed. A, B, and D contain four children each. This saves parsing 18 nodes, while only six embeddings need to be loaded from cache or recomputed.

## B ROUNDED REGRESSION ANALYSIS

Since we initially wanted to predict the number of splits and merges associated with a given tree and operation, we can also evaluate our model prediction, rounded to the nearest integer, against the actual values. Figure 8 shows that despite our model being optimized with an MSE loss, we still perform very well on this categorical task and achieve an F1 macro and micro score of 0.57 and 0.91, respectively. If we were to clamp the model output to 3, similar to a normal categorical setup, the F1 macro score would improve to 0.72.

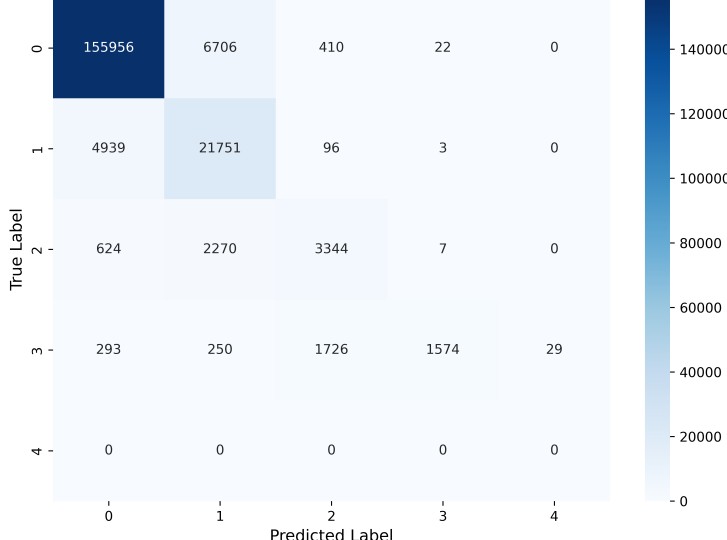

Figure 8: The confusion matrix of our rounded regression experiment highlights satisfactory performance, despite the model predicting a higher cost than 3.5 in some instances.

## C  POLICY AND REWARD DISTRIBUTION

The following two graphs illustrate the space of possible orderings for one fixed tree.

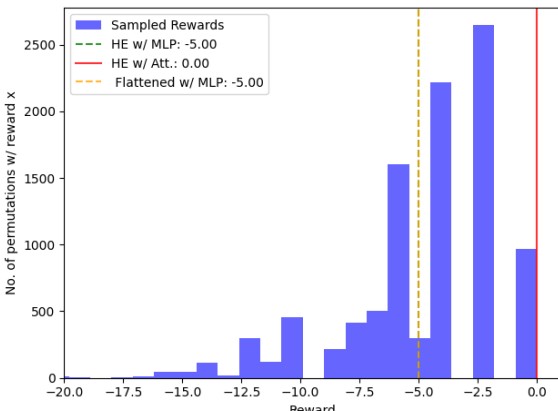

Figure 9: The summed-up received the reward of our best-performing model, red, as well as the flattened input alternative. In comparison, we plot the resulting reward for 1 000 permutations of the 12 executed actions. In this case (seed: 98476), the agent performed exceptionally well when compared to the distribution of possible rewards. A detailed analysis of the input showed no reliable indicator of why this was caused. This quantile position was the foundation of Figure 6. A comparison of this setup and the following one is provided in the caption of Figure 10.

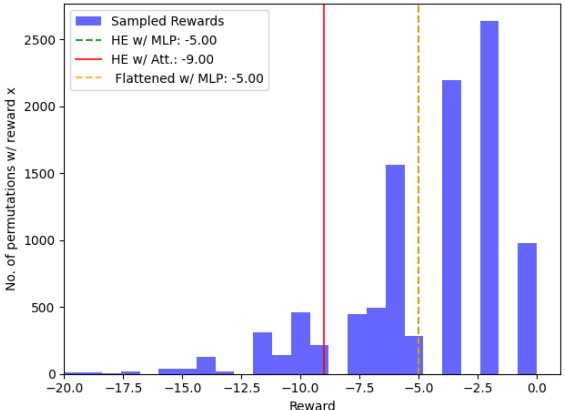

Figure 10: The summed-up received the reward of our best-performing model, red, as well as the flattened input alternative. In comparison, we plot the resulting reward for 1 000 permutations of the 12 executed actions. In this case (seed: 50010), the agent performed exceptionally poorly when compared to the distribution of possible rewards. It seems that the tree structure is at fault, even though there are only marginal differences compared to the tree that formed the initial state in the episode in Figure 9. In this case, the insertion of one value caused two splits and the addition of a full layer to the tree. A costly operation, which could have been avoided but was not properly predicted by the agent. This shows that even small inaccuracies can cause costly operations.

## D BASELINE REWARDS

Table 1 displays the average reward of all evaluated heuristics.

| Heuristic | Avg. Reward (over 10 000 episodes) |
|---|---|
| Alternating | **-4.28** |
| Inserts First | -6.44 |
| Deletes First | -8.70 |
| Random | -4.76 |

Table 1: The average reward of all four described heuristics. Interestingly, deleting the elements first causes more reconstructions due to underfilled nodes than randomly inserting. The same goes for executing all inserts first. Random performs worse than alternating.

## E HYPERPARAMETERS

The following table includes the hyperparameters used to conduct the experiments.

| MDP | |
|---|---|
| Gamma | 0.999 |
| **Optimizer** | |
| Step-Size | 0.1 |
| **Architecture** | |
| Embedding Size | 64 |
| Num. Feedforward Layers | 2 |
| Num. Attention Heads | 4 |
| Activation Function | ReLu |
| Batch Size | 512 |

Table 2: Excerpt of parameter choices for the learning setup as well as the optimizer.

Table 3 presents our tree parameter choices for the experiments.

| Parameter | Value |
|---|---|
| Node-Size $b$ | 4 |
| Elements in Tree | 24 |
| Number of Insert Ops at $s_0$ | 6 |
| Number of Delete Ops at $s_0$ | 6 |

Table 3: Environment variables for the experiments.

## F    TRAINABLE PARAMETERS

Our model scales with larger tree sizes without having to increase its parameter size, which is a fundamental improvement over other architectures.

| Values in Tree | Architecture | Number of Parameters |
|---|---|---|
| 24 | HE w/ Att. | 47,302 |
|  | Flattened | 54,272 |
| 124 | HE w/ Att. | 47,302 |
|  | Flattened | 246,272 |
| 1,024 | HE w/ Att. | 47,302 |
|  | Flattened | 6,006,272 |

Table 4: Number of parameters required for the two compared architectures across increasing tree sizes.

## G    STORAGE SPACE EXPERIMENT

The experiment setup consisted of a randomly generated tree, which subsequently was filled with six additional elements, while six elements were eventually removed. The resulting tree was then reused to add another six elements while deleting another six. This process was repeated ten times. The whole experiment was repeated ten times to reduce the influence of randomness on the measured results. We compare the average number of nodes required in this setup with the results achieved by the random heuristic. To ensure equality, the process was initialized with an equal starting tree and the same inputs. Figure 11 illustrates the results. It displays the number of nodes used by the agent to store the data available a step $t$, a lower number means the same data is stored in fewer nodes and thus the relative usage is better.

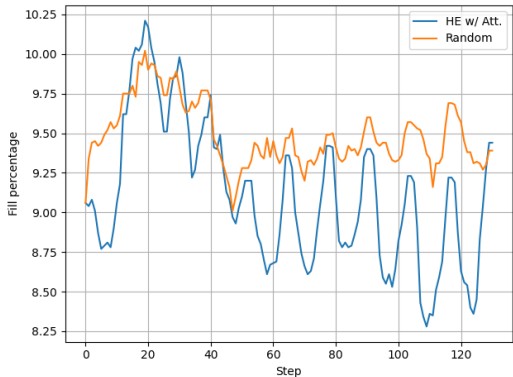

Figure 11: The average number of nodes in the graph for the experiment described in Section 6.3.

## H    INFERENCE TIME AND HARDWARE

For this system to be practically applicable, it is necessary that the inference is reasonably fast. Table 5 compares the time in milliseconds required for one forward pass on the hardware used to perform the experiments. It shows that a single forward pass, without caching, can be performed in a few milliseconds. In this scenario, all data read is already in main memory. Thus, it is assumed that the embeddings are readily available. In this system, a forward pass is fast enough to be worth the computation time when 55%. All of our experiments were conducted using either a single Nvidia A40 GPU or consumer-grade hardware. The RL training runs lasted the longest, at 6–8 hours per run, depending on the model configuration, while the regression experiments lasted less than 2 hours.

| # Values per Tree | Runtime (ms) |
|---|---|
| 24 | 1.89 |
| 124 | 2.42 |
| 1,024 | 7.21 |

Table 5: The average execution time for a forward pass in all attention-based setups. It grows in logarithmic time.

## I    OBSERVED POLICIES WITH HIDDEN TREES

The agent is able to learn well-performing policies even if it does not have access to the tree itself. Figure 12 illustrates the learning process.

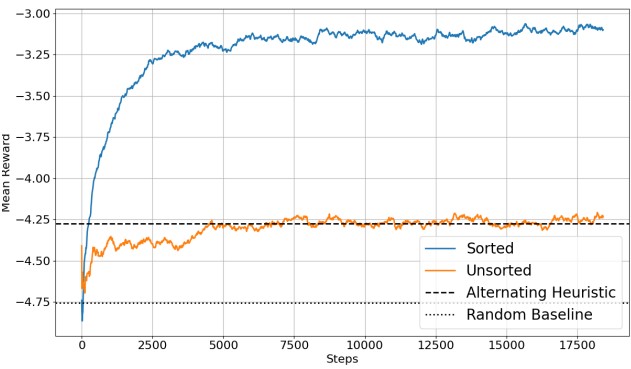

Figure 12: The learning curve of two agents that only receive the available operations and input, without ever observing the tree. While unsorted inputs lead to a result on par with randomly executing actions, sorted inputs lead to a policy that still outperforms any evaluated heuristic noticeably while only requiring a very small model and no tree reads at all.

The following figure displays the execution order chosen by the agent, which only observes the actions in 40 episodes with 12 steps each. It shows that there is no obvious policy that we could reproduce with a rule-based approach. It reliably outperforms each of the heuristics designed and evaluated by us.

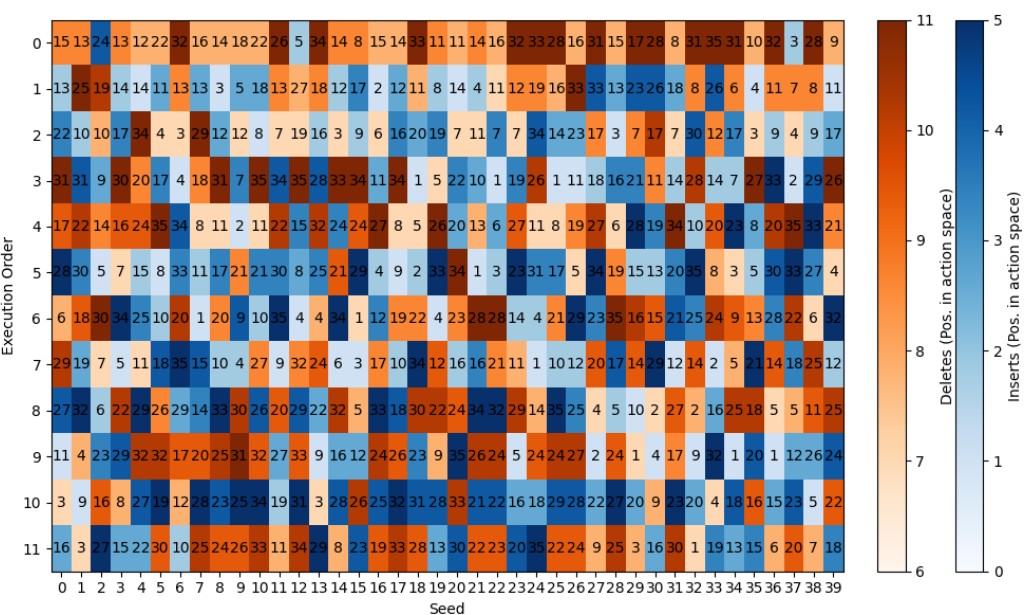

Figure 13: Each column shows the actions for one episode. The color represents whether it is an insert (blue) or a delete (red), so the position of the value in the action space, not in the space of possible values, with which each action can be associated. The hue shows the size of the value relative to the others, also encoded in the position in the sorted action. The agent tends to perform inserts and deletes in an alternating fashion, but also stacks multiple operations of the same type from time to time. The inserted values do not always resemble the deleted values, and values in close proximity do not always follow each other. Both patterns can be observed, though. For example, in seed number 29, 15 gets inserted, then 16 and 14 get deleted, then 10 gets inserted.

