# OpenReview forum: "Optimizing Operations on B-Trees Using Proximal Policy Optimization and Hierarchical Attention-Based Models"
_ICLR.cc/2026/Conference — Submitted to ICLR 2026_

### Official Review · Reviewer_PV9B · 2025-10-23

**Soundness:** 2
**Presentation:** 1
**Contribution:** 2
**Rating:** 2
**Confidence:** 5

**Summary:**

The paper proposes a reinforcement learning framework to optimize the ordering of insert and delete operations on B-trees to minimize write costs. The authors introduce a hierarchical attention-based neural network architecture that recursively embeds B-tree structures and uses PPO to learn an operation scheduling policy. The approach includes a caching mechanism to reduce computational overhead during inference. Experiments on small B-trees (24 values, b=4) demonstrate 55% improvement over random execution and outperformance of simple heuristics.

**Strengths:**

**S1**- Novel Problem Formulation: The paper addresses a practical problem—optimizing batch operations on B-trees—from an RL perspective, which is relatively unexplored.

**S2**- Practical Caching Mechanism: The invalidation-based caching system is well-motivated for real-world deployment, recognizing that most operations affect only local tree regions.

**S3**- Open-Source Commitment: The authors promise to release their simulation toolkit.

**Weaknesses:**

**Major Issues**

**W1**-	The core premise is questionable: Do real database systems actually benefit from reordering buffered operations? The paper provides no empirical evidences from actual database benchmarks (for example TPC-C, YCSB).

**W2**-	 All experiments use small trees (24 values, b=4). Real-world B-trees contain millions of entries with b≥100. Also, in general the b-tree systems operation is between 100-500 ns, and in worse case goes to micro-seconds but the GPU dispatching and model forward is in order of milliseconds. B+tree by satelliting data improve the speed evenmore.

**W3**-	Modern databases already use sophisticated buffering and write-ahead logging. The assumption that operations can be buffered and reordered contradicts transactional consistency requirements (ACID property) in most database systems or MVCC systems where operation order affects correctness. No comparison with existing database optimization techniques like bulk loading, LSM-trees or B\epsilon-trees which are designed specifically for write optimization.

**W4**- The "baselines" (alternating, insert-first, delete-first) appear arbitrary and are not motivated by database literature. No comparison with existing B-tree bulk operation algorithms from database literature (for example Jannink 1995's or Learned indexes (Kraska et al. 2018) works cited but not compared or Bulk loading algorithms). Missing comparison with sorted insertion order, which is known to be optimal for many B-tree scenarios.

**W5**- Equal numbers of inserts and deletes (6 each) is not representative of real workloads. Random tree initialization may not reflect realistic data distributions. Reward function (Equation 3) weights are not justified: why is $p\ne q$ for leaf vs internal operations? Gamma=0.999 means agent heavily discounts future rewards, contradicting the claim that operation order matters equally. Moreover, the implementation on python is not a perfect development tool for systems specially for time sensitive designs.

**W6**- Regression experiments (Section 4) predict cost on pre-generated trees but dont validate if this prediction ability translates to better scheduling. The "no tree" agent achieves -3.1 average reward, only 29% worse than the full model (-2.17), suggesting tree information provides marginal benefit. No ablation study on critical design choices (e.g., number of attention heads, embedding dimension).

**Minor Issues**

**W7**-	 Inference time measurements(Table 5) only considered for forward pass, ignoring tree reading costs which the paper claims are expensive. Storage space improvements (Section 6.3, Figure11) are temporary. No wall-clock time comparison showing net benefit after accounting for inference overhead.

**W8**-	Statistical Rigor: Figure 5 shows variance without confidence intervals. Also, only 1,000 trees tested for quantile analysis (Figure 6) insufficient for robust conclusions.

**W9**-	Presentation Issues: Some notations presented without clear definitions (like p, q in Equation 3 introduced without values). Figures 9 and 10 show chery-picked examples without systematic analysis of failure modes. Missing related works on write-optimized data structures beyond brief mentions.

**Questions:**

Addressing W1,W2, W4,W5

**Q1**- How does your approach integrate with transaction management, concurrency control, and crash recovery in actual database systems?

**Q2**- $10^8$ episodes over 6-8 hours—how does this training cost amortize in practice? How frequently must models be retrained as data distributions change?

**Q3**- **Failure analysis**: Figures 9 and 10 show dramatic performance variance. Can you characterize when and why the agent fails?

---

### Official Review · Reviewer_p98N · 2025-10-24

**Soundness:** 2
**Presentation:** 2
**Contribution:** 2
**Rating:** 4
**Confidence:** 3

**Summary:**

The paper highlights a key challenge in B-trees: insertions may cause node overflows requiring splits, while deletions may lead to node underfill and trigger merges. Both operations can induce local or even cross-level structural modifications, which are costly and complex.

To address this, the authors propose a tree representation that can be efficiently updated during modifications and leverage it to automatically train an agent to determine the optimal execution order. They adopt PPO as the optimizer and employ a hierarchical attention encoder to efficiently embed large hierarchical datasets within the tree structure.

**Strengths:**

1. The paper clearly identifies the main challenges of B-trees and proposes an effective solution.

2. The introduced hierarchical parallel computing and caching mechanism is highly valuable for reproducibility and future research.

**Weaknesses:**

Overall, the paper studies B-trees, a topic more relevant to database research (e.g., SIGMOD, VLDB, ICDE), which may not attract significant interest from the ICLR audience.

1. In Chapter 3.1, although the hierarchical model architecture is described, providing examples or figures illustrating specific steps would help enhance the reader’s understanding. The methodology section currently lacks formulas and visual aids to support the explanation.

2. The overall structure of the paper could be improved. I suggest combining Chapter 4 and Chapter 5, with clear explanations, so that readers can better understand the connection between the regression experiments and the reward function.

3. Including the pseudocode for PPO in Chapter 5 would further clarify the methodology.

4. The paper mainly compares the proposed approach with random order and several hand-crafted heuristic methods, but lacks comparisons with state-of-the-art strategies in real systems or established database baselines. It is therefore difficult to demonstrate that the proposed method outperforms mature engineering practices.

**Questions:**

Please address the weaknesses.

---

### Official Review · Reviewer_TKb3 · 2025-11-04

**Soundness:** 2
**Presentation:** 2
**Contribution:** 2
**Rating:** 4
**Confidence:** 3

**Summary:**

This paper proposes to optimize B-tree insert and delete operations by learning an execution order through a reinforcement learning framework (PPO) combined with a hierarchical attention-based encoder. The goal is to minimize costly node splits and merges when B-trees are stored on slow persistent storage. Empirical results demonstrate that the learned policy outperforms several heuristic baselines and achieves up to 55% improvement over random operation ordering on small-scale B-trees.

**Strengths:**

1. This paper focuses on applying reinforcement learning method  (PPO)  to fine-grained B-tree operations, which is interesting.
2. Reducing write cost and rebalancing overhead in persistent storage is a relevant systems problem.

**Weaknesses:**

- The introduction of RL methods appears contrived  and fails to demonstrate domain knowledge modeling of the index structure. The so-called “hierarchical attention encoder” is essentially similar to the normal recurrent transformer, lacking novelty.

- How scalable is the hierarchical encoder—what is the computational cost with respect to tree size and branching factor?

- The reported “reward” does not necessarily translate to true performance gains. The claimed improvement  (approximately 55% better than random operations) was measured in a simplified synthetic environment using a manual cost function—How about actual runtime metrics in a controlled setting? Show similar results would significantly enhance the credibility of this paper.

- Baselines in this paper is overly simple; while claiming scalability to larger tree structures, no convincing proof is provided (while the paper claims scalability to larger B-trees, the results show a sharp performance degradation as tree size increases, with no convincing analysis or mitigation strategy).

**Questions:**

As stated above.

---

### Meta-Review · Area_Chair_KEgX · 2025-12-07

**Summary:**

Reviewers raised substantial concerns about weak novelty, unrealistic assumptions, limited scalability, and inadequate experimental validation. They found the RL formulation contrived, the hierarchical encoder insufficiently justified, and the evaluation restricted to tiny synthetic B-trees with overly simple baselines. Practical relevance to real database systems and performance claims remained unconvincing.

**Reviewer Concerns:**

It appears that the authors did not provide a rebuttal. So the key outstanding issues remain: (1) lack of real-system validation, realistic workloads, or comparisons with database baselines; (2) poor scalability and questionable practicality given transactional constraints; (3) weak novelty and limited technical justification; and (4) major methodological gaps and unaddressed experimental weaknesses.

**Reviewer Scores:**

PV9B: 2
p98N: 4
TKb3: 4

---

### Decision · Program_Chairs · 2026-01-26

Reject